# Old Timber Reinforcement with FRPs

**DOI:** 10.3390/ma12244197

**Published:** 2019-12-13

**Authors:** Janusz Brol, Agnieszka Wdowiak-Postulak

**Affiliations:** 1Faculty of Civil Engineering, Silesian University of Technology, 44-100 Gliwice, Poland; Janusz.Brol@polsl.pl; 2Faculty of Civil Engineering and Architecture, Kielce University of Technology, 25-314 Kielce, Poland

**Keywords:** old wood, bending strength, load capacity, reinforcement, repair, carbon-fibre mats, aramid-fibre mats, basalt-fibre rods, glass-fibre rods, carbon-fibre strips

## Abstract

The tests included the determination of the reinforcement effectiveness of old larch timber originating from a building built in 1860 with the use of carbon-fibre and aramid-fibre mats and strips, BFRPs and GFRPs. The test results showed that in old solid timber pieces from European larch (*Larix decidua* Mill.), the highest mean flexural bending capacity occurred in samples reinforced with carbon-fibre mats (increase in flexural bending capacity was 60.66% in relation to non-reinforced elements), while the lowest flexural bending capacity of the tested components occurred with reinforcement with GFRP (10 mm in diameter) (increase by only 19.04% in relation to non-reinforced elements). Additionally, bending tests of repaired 130-year-old pine (*Pinus sylvestris* L.) beams were shown (real-size scale) using CFRP strips and mats. The problems associated with the delamination of the CFRP strip due to uneven deformation of the damaged timber surface and the effectiveness of these repairs are also shown.

## 1. Introduction

In addition to stone and earth, wood is the oldest building material. It is still used today for building structures. Wood has advantageous physical and chemical properties, high strength, and low specific weight. This material is also resistant to the action of many chemical substances. Unlike many plastics, wood maintains a chemical balance with the environment. Under appropriate moisture conditions, structural components made of timber have a long lifespan at moderate maintenance costs during the structure’s service life. Timber undoubtedly also has many drawbacks related to its structure (e.g., knots, shakes, twisted fibres, non-uniform anisotropy) and susceptibility to biological corrosion. As a result, constant care and periodic checks are required for timber structures to remain in service. Inspections, carried out on a regular basis, make it possible to reveal harmful destructive processes that can result in the deterioration of the structure serviceability properties, or failures. When the performance parameters of timber components decline, it may be necessary to reinforce or repair them [1,2,3,4,5,6,7,8], or in extreme cases, replace them. Replacement of existing elements with new ones is often impossible or very difficult, therefore structural members are usually repaired or reinforced.

The following methods of wooden beam reinforcement can be distinguished [1,5,9]:structural protection and reinforcement by introducing extra components (independent structural reinforcement, suspended structural reinforcement system or reinforcement cooperating with the existing structural system), andstructural operations (increasing the technical and operational values of damaged components).Usually, reinforcements working with an existing structural system include [1]:timber beams reinforcement along their entire length by joining the side surfaces with reinforcing components such as planks or log tops, steel sections and the introduction of reinforcement into the beam tension zone;reinforcement on a beam section, including the replacement of the damaged beam section with a supplementary element, the removal of the damaged beam end and the floor beam suspension by steel stirrups;reinforcement based on the floor beam-reinforced concrete slab composition, andbeam reinforcement with tie rods.

Structural treatments include surface treatment (protection of timber against biological corrosion), fire protection, deep pressure impregnation, injection treatments (strengthening of wood structure or enhancement of its cross-section by means of mixtures (injection grouts) based on epoxy resins.

In the 1970s, there was an increase in the methods of timber structure reinforcement as a result of developments in the field of epoxy adhesives used for fixing reinforcing elements. The reinforcement of timber, reinforced concrete, masonry and steel structures with the fibrous composites is very popular at present [10,11,12]. Self-curing composites based on epoxy resins are available on the market, enabling a significant increase in load capacity [1,2,5]. The favourable use of fibre-reinforced plastic composites is a result of [2,5,13]: its low bulk density, corrosion resistance, high fatigue strength, high tensile strength, easy assembly, heat flow resistance, and stray current formation.

Composites are produced on the basis of glass fibres (usually GFRP/glass fibre reinforced polymers), aramid fibres (AFRP/aramid fibre reinforced polymers), carbon fibres (CFRP/carbon fibre reinforced polymers), and hybrid materials (consisting of different types of fibres, e.g., carbon fibres and glass fibres) [13,14]. Glass fibres show poor resistance to alkaline environments and lower fatigue strength. Aramid fibres have similar strength parameters to glass fibres, but are more resistant to fatigue [13]. Due to the fact that carbon fibres are characterized by higher stiffness than glass and aramid fibres, and high resistance to most aggressive chemical agents as well as high temperatures, these fibres have the best properties among fibres used in the production of fibre composites [13]. FRP fibre composites are used to produce strips, mats, reinforcing rods (smooth, deformed), prestressing ties (prestressing rods, prestressing ropes), structural elements, such as sections, trusses and slab elements.

If it is not possible to perform the reinforcement in the traditional way, and if there is a need for quick and effective repair of a structure in a pre-failure condition, the use of fibrous composites is an effective way. The low weight and easy concealability in the cross-section allows them to be used for the rehabilitation and repair of structures in existing buildings.

Concepts based on the use of fibrous composites, such as carbon, aramid, glass, and recently basalt fibres for the reinforcement of timber structures corresponds to a popular and widely applicable method that involves steel elements (rods, metal sheets, flat bars) glued to the surface of the external timber elements or glued into their cross-sections. The effectiveness of the bonding of different materials depends, to a large extent, on the similarity of their physical and mechanical properties. The fibrous structure of the composites resembles that of timber. In [15], timber is considered to be an example of a natural composite, and consequently a “composite is reinforced with another composite”.

Reinforcement of timber elements with fibrous composites by gluing tapes to the external surfaces of the reinforced element also resembles similar solutions commonly applied to reinforced concrete structures. The latter techniques have been well investigated.

The reinforcement of timber structures with composites has not yet become a common practice. The advantages and disadvantages of such solutions have not been thoroughly examined. This applies especially to timber reinforcement in structures that are more than a hundred years old. Another problem is the examination of the effectiveness of larch timber (*Larix decidua* Mill.) reinforcement with FRPs. Therefore, this kind of timber is focused on in this study. As larch timber shows high durability, it has been a welcome option for religious or prestigious buildings, and also those exposed to disadvantageous environmental impacts. A good example of a long service lifespan of larch timber outdoor structures is provided by a historic telecommunications tower located in Gliwice [16]. The tower, built in 1935, is the highest facility of this type.

Chapter 2 of the study discusses the results of tests on the effectiveness of bending reinforcement of old larch timber obtained from the over 150-year-old facility. In the tests, the analyses examined the influence produced by different composite materials on the reinforcement effect and the failure mode of old timber without visible damages. The tests aimed to estimate the reinforcement effect, and the possibility of improving the structural performance, as well as that of the elements of existing structures. The investigations constituted a part of the research performed for the doctoral dissertation of one of the authors [6].

In Chapter 3, two methods of repair of individual old damaged timer beams are discussed. The over 130-year-old pine beams were repaired using CFRP tapes and mats, depending on the type of damage. It was demonstrated that the method of repair depends on the type of damage suffered by the elements. In the tests reported in the literature, damage is very often simulated, which does not always represent the actual (natural) damage characteristics. As shown in Chapter 3, the classical approach may be inappropriate, and should be slightly modified. The repairs aimed at restoring the original capacity of the beams. The tests of the proposed repairs were performed on full-scale elements.

## 2. Testing on Small Samples

### 2.1. Research Materials

The tests included the determination of the physical and mechanical properties of the European larch (*Larix decidua* Mill.) originating from a building built in 1860. The test components were cut from the old floor beams located in a brickyard in Machory, Poland (see Figure 1).

The measurements, performed according to PN-EN 408+A1:2012 [17], with an accuracy of 1%, were taken in the laboratory immediately before destructive tests. All measurements were performed on conditioned samples collected from the midspan of the test piece. The relative air humidity was 60–65%, and the air temperature approx. 20–22 °C. The following fibrous composites were used in the tests [2,3,6,7,8]:

• Coal-fibre mats

To reinforce the beams, carbon-fibre mats were used, which were manufactured on the basis of T700S carbon fibres (density 1.8 g/cm^3^, tensile strength 4900 MPa, modulus of elasticity at 230 GPa).

• Aramid-fibre mats

To reinforce the beams, aramid-fibre mats were used, which were manufactured on the basis of Kevlar 49 DuPont TM (density 1.44 g/cm^3^, tensile strength 3600 MPa, modulus of elasticity at 124 GPa).

• Adhesive

The slats were joined with D4 PVAc adhesive (density approx. 1.10 g/cm^3^, viscosity 13,000 mPa.s) and epoxy adhesive. The adhesive layer based on epoxy resin was obtained by mixing LG 385 epoxy resin (density 1.18 ÷ 1.23 g/cm^3^, viscosity 600 ÷ 900 mPa.s) with HG 385 hardener (density 0.94 g/cm^3^, viscosity 50 ÷ 100 mPa.s). After mixing resin and hardener, the adhesive obtained bending strength of 110 ÷ 120 MPa and modulus of elasticity of 2700 ÷ 3300 MPa. The epoxy adhesive (LG 385, HG 385) consisted of two parts, which were mixed in a 2:1 ratio by volume, in accordance with the product standard requirements.

• BFRPs and GFRPs

The elastic modulus values and final strains of the GFRPs and BFRPs are shown in Table 1.

Figure 2 illustrates the old pieces reinforced with carbon-fibre and aramid-fibre mats, as well as BFRP and GFRP (10 mm in diameter) (30 × 30 × 600 mm).

### 2.2. Research Methodology

The tests were performed on samples conditioned in air of (20 ± 2) °C and relative humidity (65 ± 5)%. The samples were conditioned to a constant mass and stored in test rooms (24 h), stacked and tightly wrapped in a vapour-proof material.

The bending strength test was assessed on the basis of PN-EN 408+A1:2012 [17] using ±250 kN electromechanical testing machine (Zwick, Wrocław, Poland) (see Figure 3). The tested samples were symmetrically loaded with two concentrated forces, with a span equal to 18 times the cross-sectional height of the beam. In the test, the span between the supports was 540 mm. The load was applied at a constant speed of travel so that the maximum load was reached after (300 ± 120) s. The load speed of the sample was determined by preliminary tests. The optimum time to reach F_max_ was 300 s.

The bending strength was determined on the basis of PN-EN 408+A1:2012 [17], according to the formula:(1)$$f_{m}=\frac{3Fa}{bh^{2}}$$
where: $f_{m}$ is the bending strength (N/mm^2^); F is the load [N]; a is the distance of the force application point from the nearest support (mm); b is the piece width in the bending test or the dimension of the smaller side of the section (mm); h is the piece height in the bending test or the dimension of the greater side of the section (mm).

### 2.3. Test Results on Small Samples

Wooden samples (including non-reinforced—SZL; reinforced with carbon-fibre mats—WSZLw; reinforced with aramid-fibre mats—WSZLa; reinforced with BFRP—SWZLPb; reinforced with GFRP—SWZLPsz), were visually strength sorted according to PN-D-94021:2013-10 [18] and divided into three quality classes: KW—exclusive class, KS—medium quality class, KG—lower quality class, and designated for “rejection” (see Table 2) [19,20,21,22,23].

The loading force value and the image of the failure of the test piece were recorded during the test. The load speed in this test was 0.06 m/s.

The results of the bending tests of solid timber samples with defects, European larch *Larix decidua* Mill. from a building built in 1860 (30 × 30 × 600 mm), are shown in Table 3.

Bending strength of solid timber samples with defects, European larch (*Larix decidua* Mill.) from a building built in 1860 (30 × 30 × 600 mm) reinforced with carbon-fibre and aramid-fibre mats, BFRPs and GFRPs (10 mm in diameter), are shown in Table 4. The load speed in this test was 0.073 mm/s.

Reinforcement effectiveness of timber pieces with defects of the European larch (*Larix decidua* Mill.) originating from a building built in 1860 (30 × 30 × 600 mm) using carbon-fibre and aramid-fibre mats, BFRPs and GFRPs (10 mm in diameter), is shown in Figure 4.

## 3. Real-Size Scale Tests on Beams

### 3.1. Research Materials and Research Methodology

With 130-year-old timber beams (pine) (*Pinus sylvestris* L.), which were previously tested for bending [24], it was decided to try to repair them and determine the effectiveness of this repair. The average dimension of the tested beams and the support spacing during the test are given in Table 5.

The bending tests were performed at a support spacing of l = 4.5 m. The test diagram and a view of the test stand are shown in Figure 5. Force, deflection and deformation on the bottom and top of the beam were measured during the tests.

The aim of this test was to determine the possibility of reinforcement of the historic beams. The damage was actual, not simulated, as was the case in various tests reported in [25,26]. Two timber beams, marked BD5 and BD6 in previous tests [24], were selected. The beams did not show signs of biological corrosion, but the traces of insect feeding could be found.These beams were slightly broken beforehand during the destructive bending tests. The nature of the damage, however, made it possible to repair them. This corresponded to a frequently found potential failure condition of the floor beams; therefore, it was decided that repairs should be made. For this purpose, S512 carbon-fibre (modulus of elasticity 165 GPa, field 60 mm^2^) and M1214 carbon-fibre (modulus of elasticity 240 GPa, field 168 mm^2^) were used. It was decided that the beams would be repaired by gluing carbon-fibre strips to the lower surface of the beams with SikaDur30 epoxy adhesive, after cleaning these surfaces. The method of repair depends on the nature of the damage.

### 3.2. Test Results for Wooden Beams

The results are shown in Table 5 and Figure 6, and are described in more detail below. The BD5 beam was repaired with a S512 strip by gluing it to the lower surface of the beam over its entire length (Figure 7). The reason for reinforcement over the entire length was the minor damage to the fibres in the range from 1/3 to 2/3 of the beam span. The main damage to the BD5 beam, on the other hand, occurred close to the centre of the beam span and consisted of a rupture of the lower timber fibres. This solution enabled restoration to 80% of the original load-bearing capacity of the beams, simultaneously maintaining the stiffness of the original beam. With regard to the permissible deflections of l/300, the rigidity practically remained unchanged. Only when the deflection values were higher did the original damage to the beam become apparent. The strip was damaged by loosening in the anchorage zone (Figure 8).

In case of the BD6 beam, it was decided to glue an M1214 strip with a 240 MPa module for 1/3 to 3/3 of the beam length (Figure 9) due to the nature of the damage to the beam. In this beam, the damage was deeper, and occurred at around 2/3 of the beam span (under a concentrated force from earlier tests) and covered half of the section height with a simultaneous longitudinal crack.

Due to longitudinal cracks in the middle of the beam height, uneven displacements at the adjacent edges of the cracks were visible during the test (Figure 10), and fast delamination occurred at the timber-CFRP strip joint. As a result, only about 58% of the primary load capacity was achieved while maintaining the beam stiffness. With regard to the permissible deflections l/300, only a small loss of stiffness was observed.

Due to the destruction form of the timber-M1214 strip joint as a result of delamination, it was decided to repeat this test on the same beam while the joint was reinforced at the site of the damage to the beam by additional wrapping of the spot with a carbon-fibre mat (Figure 11 and Figure 12). Wrapping the beam in the damaged area with the mat was intended to prevent independent vertical movements on both sides of the contact, and thus to prevent delamination. As shown in Figure 6, this solution significantly improved the repair result and recreated 90% of the original load capacity. With regard to the permissible deflections l/300, a small loss of stiffness was also observed.

## 4. Conclusions

On the basis of the tests performed, it was found that:Reinforcement of timber structures with the use of fibrous composites is relatively easy to perform, effective and resistant to external factors, which was also shown in [26,27]. These solutions should especially be used where traditional reinforcement is impossible or inappropriate for some reason.The reinforcement effectiveness depends on the fibrous composites used, and their quantitative content. Based on [26] and other research conducted by the authors [6,27], it was found that the reinforcement effect is also related to the timber species. The authors’ investigations demonstrated that the effectiveness of larch timber reinforcement is higher than that of pine timber.It should be added that reinforcement with carbon-fibre strips or mats does not practically increase the size of the cross-section, and it is easily possible to cover the reinforcing pieces or place them in the cross-section in the case of historic buildings. The use of CFRP to reinforce timber structures, especially along the fibres, also has the advantage of having a comparable coefficient of thermal expansion as timber.Some structural and geometric features reduced the bending strength of non-reinforced pieces, especially occurring grain slopes, cracks, decay, etc.The knots had a negative impact on the bending strength of the timber. Sapstain did not affect the mechanical properties of non-reinforced and reinforced pieces.Static bending strength increased with increasing timber density and parallel fibre system. The highest strength was observed in the case of timber with a fibre course as close as possible to the direction of the sample axis.Higher strength values were obtained for pieces with higher shares of heartwood and summer wood present.The applied reinforcement positively affected the work of pieces with timber defects, making it possible to obtain higher reinforcement effects, especially in the case of sawn timber of lower classes—KS, KG.The damage of pieces during the bending test was mainly due to the folding of the compressed plane and the tearing of the tension plane. In the reinforced pieces, composite mats were torn off during bending strength tests.In solid timber pieces of European larch (*Larix decidua* Mill.) originating from a building built in 1860, the highest average bending load capacity occurred when samples were reinforced with carbon-fibre mats (an increase of about 61% compared to non-reinforced pieces), and the lowest when samples were reinforced with 10-mm GFRP (an increase of 19%).Comparing the reinforcement effectiveness of solid timber pieces of the European larch (*Larix decidua* Mill.) from a building built in 1860, the increase in bending strength of a carbon-fibre mat was 60.66%, that of an aramid-fibre mat was 31.23%, BFRP (10 mm in diameter) was 20.43% and GFRP (10 mm in diameter) was 19.04%.Tests performed for the repair of damaged timber beams show that it is possible to restore their original load-bearing capacity using fibrous materials (FRP); however, it should be remembered that timber is a heterogeneous material with defects, and in the case of damage, the damage occurs irregularly. This results in uneven displacements at the adjacent edges of the crack at the next load after the reinforcement, which leads to delamination at the timber/FRP material contact (as observed in the BD6 test—repair 1). It is, therefore, necessary to perform additional anti-laminating ties (as shown in BD6 test—repair 2).

## Figures and Tables

**Figure 1 materials-12-04197-f001:**
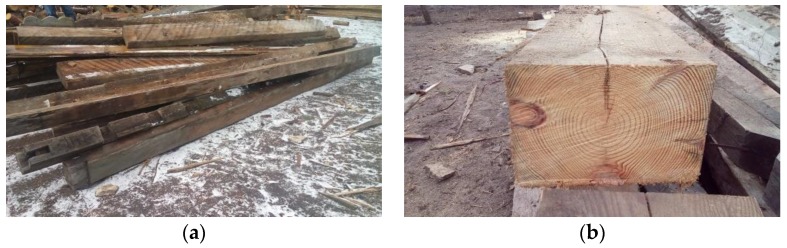
Timber components (photo by A. Wdowiak-Postulak): (**a**) floor beams; (**b**) the cross-section of the European larch (*Larix decidua* Mill.) from which the test samples were cut.

**Figure 2 materials-12-04197-f002:**
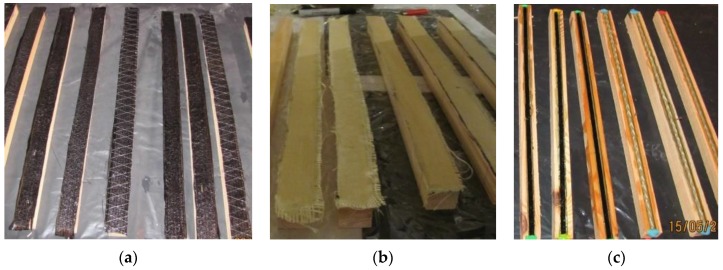
Tested pieces (photo by A. Wdowiak-Postulak): (**a**) reinforced with carbon-fibre mats; (**b**) reinforced with aramid-fibre mats; (**c**) reinforced with BFRPs and GFRPs.

**Figure 3 materials-12-04197-f003:**
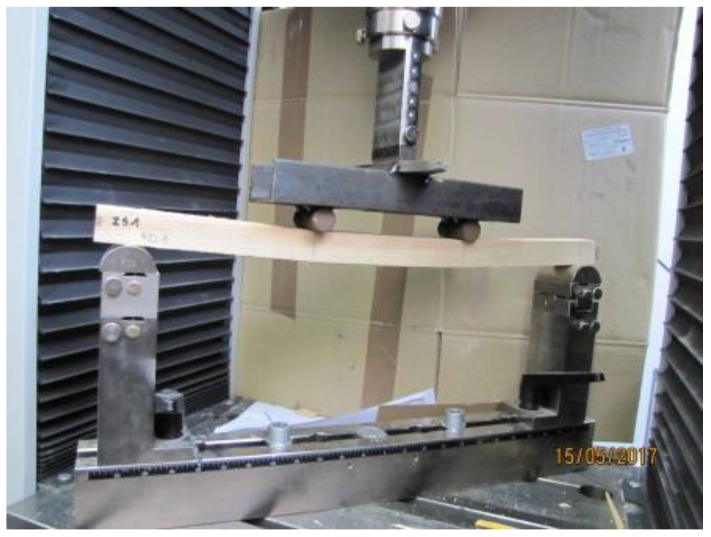
Bending strength test of timber pieces (photo by A. Wdowiak-Postulak).

**Figure 4 materials-12-04197-f004:**
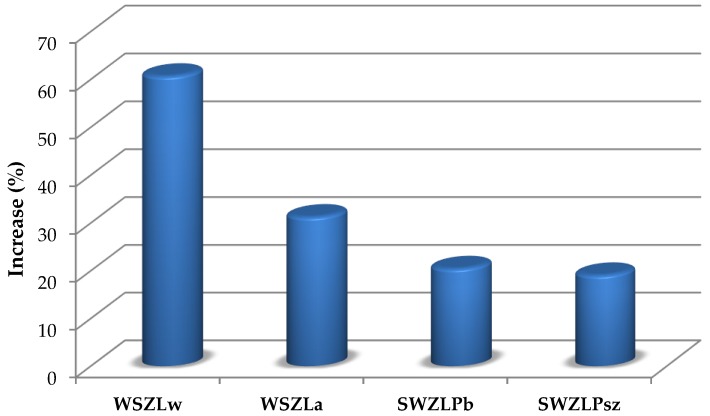
Comparison of the increase in bending strength of solid timber pieces with defects, European larch (*Larix decidua* Mill.) originating from a building erected in 1860 reinforced with carbon-fibre mats (WSZLw), aramid-fibre mats (WSZLa), BFRP (10 mm in diameter) (SWZLPb), GFRP (SWZLPsz) (10 mm in diameter) in relation to non-reinforced pieces (SZL).

**Figure 5 materials-12-04197-f005:**
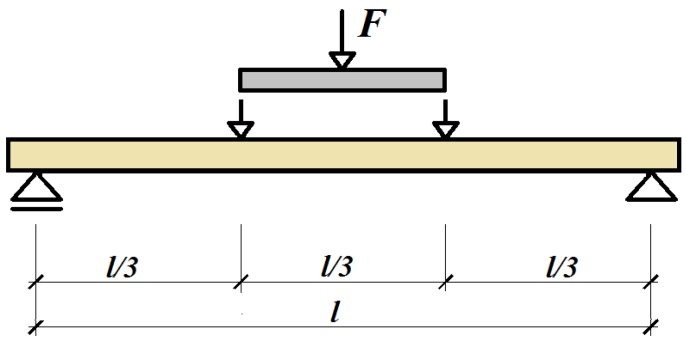
General scheme of testing.

**Figure 6 materials-12-04197-f006:**
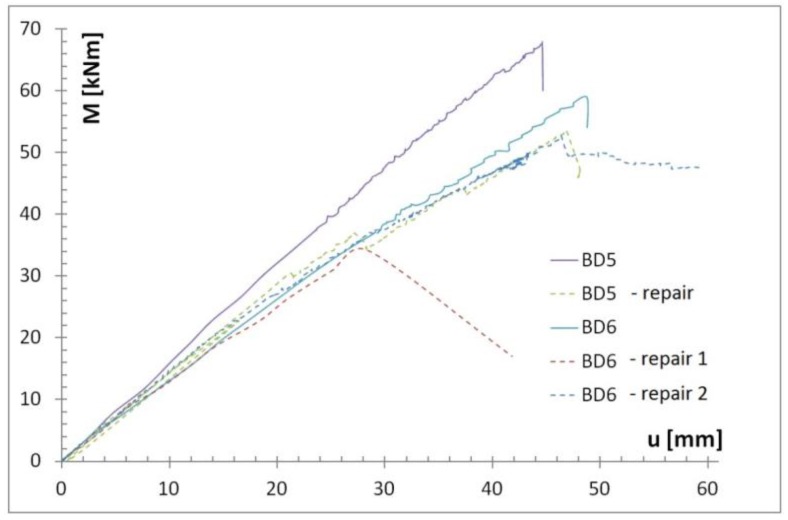
Moment—deflection dependency of tested timber beams in destructive tests and after repairing with the use of CFRP.

**Figure 7 materials-12-04197-f007:**
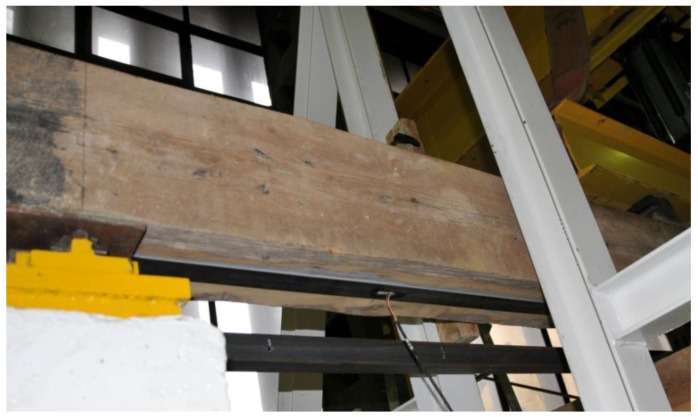
View of the beam repaired with the use of CFRP strip S512 (photo by J. Brol).

**Figure 8 materials-12-04197-f008:**
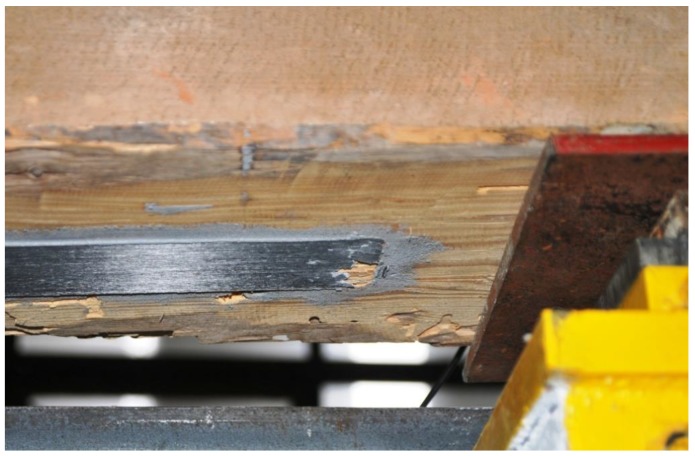
Destruction by strip delamination at the end of the anchorage zone (photo by J. Brol).

**Figure 9 materials-12-04197-f009:**
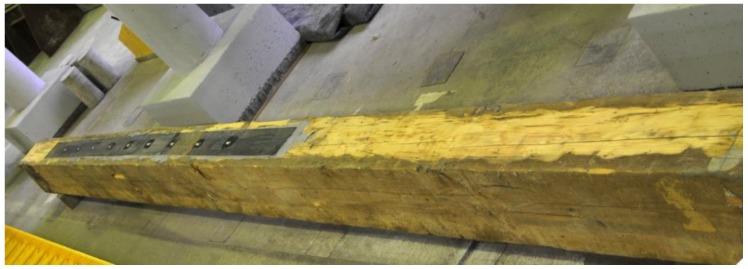
Timber beam repair with the use of CFRP strip M1214 (photo by J. Brol).

**Figure 10 materials-12-04197-f010:**
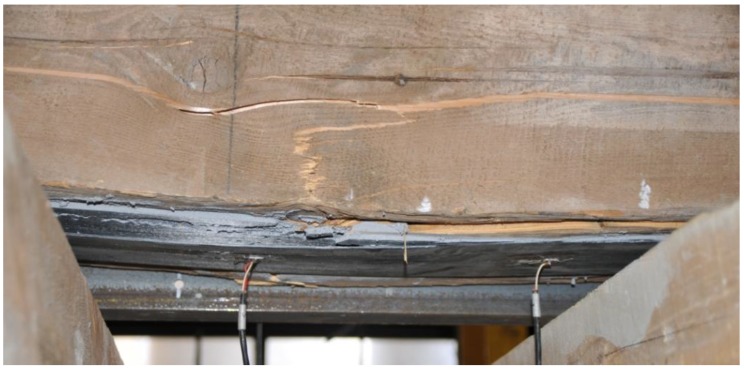
The form of destruction of the timber beam BD6 (photo by J. Brol).

**Figure 11 materials-12-04197-f011:**
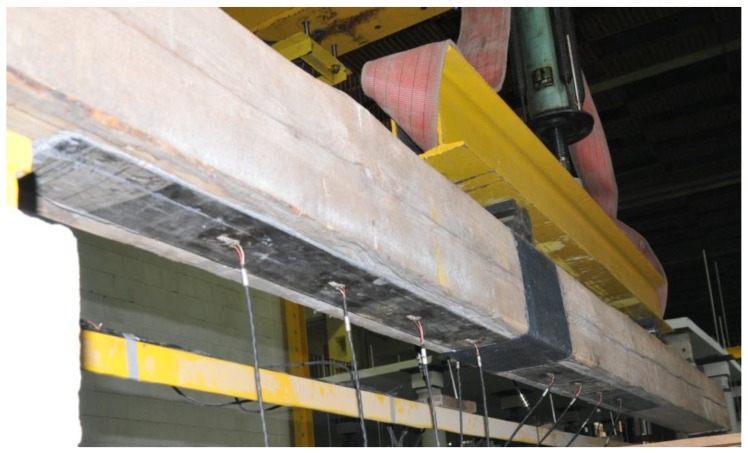
Re-repair of the timber beam with the use of CFRP strip M1214 (photo by J. Brol).

**Figure 12 materials-12-04197-f012:**
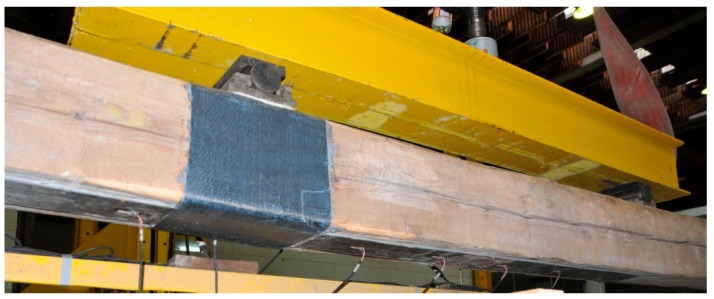
View of the additional strengthening with the use of CFRP mat (photo by J. Brol).

**Table 1 materials-12-04197-t001:** Moduli of elasticity and end deformations of FRP rods.

Symbol	GFRP	BFRP
*E*, (GPa)	55	78
*ε**_u_*, (%)	28	39

**Table 2 materials-12-04197-t002:** Characteristics of selected testing pieces (USM—marginal zone knotting index; USC—general knotting index).

Symbol	Dimension (mm × mm × mm)	Moisture Content (%)	Sample Details	Density (kg/m^3^)	Grain (mm)
SZL1	31 × 32 × 597	10.2	KW, deep sapstain, brown discolouration, sound longitudinal through knot (USM = 9/40, USC = 1/10), considerable hardwood, sapwood, early and late wood share	712.6	2.7
WSZLw1, coal-fibre mat, the European larch *Larix decidua* Mill. originating from a building from 1860.	33 × 32 × 598	9.2	KW, grain slope (0,3%), hardwood, sapwood, early and late wood share	704.7	3.6
WSZLa8, aramid-fibre mat, the European larch *Larix decidua* Mill. originating from a building from 1860.	32 × 33 × 599	9.9	KW, decaying oval knot on the plane, sound oval knot on the plane (USM = 1/5, USC = 1/10), grain slope (0,3%), burls, hardwood, sapwood, early and late wood share	673.5	2.6
SWZLPb13 BFRP 10 mm in diameter, epoxy adhesive (LG385 + HG385), the European larch *Larix decidua* Mill., originating from a building erected in 1860.	33 × 33 × 597	9.8	KW, lengthwise decayed through knot—(USM = 1/4, USC = 1/8), grain slope (0,2%), burls, hardwood, sapwood, early and late wood share	836.8	3.5
SWZLPsz16 GFRP 10 mm in diameter, epoxy adhesive (LG385 + HG385), the European larch *Larix decidua* Mill., originating from a building erected in 1860.	32 × 31 × 597	9.7	KG, sound lengthwise knot on the reinforced edge, sound oval knot on the plane—(USM = 1/3, USC = 1/4), grain slope (0.3%), heartwood, sapwood, spring early and late summer wood share	797.0	3.7

**Table 3 materials-12-04197-t003:** The results of bending tests of unreinforced solid wood samples with defects, European larch *Larix decidua* Mill. from a building built in 1860 (30 × 30 × 600 mm).

Sample Symbol	Support Spacing, (m)	Destructive Force, (kN)	The Moment of Destruction, (kNm)	Deflection (mm)
SZL1	0.54	3.94	0.71	18.3
SZL2	0.54	3.51	0.63	21.2
SZL3	0.54	4.21	0.76	26.1
SZL4	0.54	2.26	0.41	13.2
SZL5	0.54	3.01	0.54	15.8
SZL6	0.54	3.93	0.71	16.8
SZL7	0.54	3.92	0.71	19.2
SZL8	0.54	2.96	0.53	18.7
SZL9	0.54	1.83	0.33	9.3
SZL10	0.54	2.18	0.39	13.4
SZL11	0.54	3.39	0.61	20.2
SZL12	0.54	3.84	0.69	21.6
SZL13	0.54	3.59	0.65	20.8
SZL14	0.54	3.29	0.59	18.1
SZL15	0.54	2.11	0.38	10.9

**Table 4 materials-12-04197-t004:** The results of bending strength of solid wood samples with defects, European larch (*Larix decidua* Mill.) from a building built in 1860 (30 × 30 × 600 mm) using carbon-fibre mats (WSZLw), aramid-fibre mats (WSZLa), BFRP (WSZLPb) and GFRP (WSZLPsz) (10 mm in diameter).

Sample Symbol	Support Spacing, (m)	Destructive Force, (kN)	The Moment of Destruction, (kNm)	Deflection (mm)
WSZLw1	0.54	5.47	0.99	28.8
WSZLw2	0.54	4.20	0.76	29.6
WSZLw3	0.54	6.46	1.16	28.5
WSZLa7	0.54	3.87	0.70	25.9
WSZLa8	0.54	4.08	0.73	28.1
WSZLa9	0.54	4.27	0.77	26.7
SWZLPb13	0.54	4.12	0.74	30.8
SWZLPb14	0.54	4.66	0.84	21.7
SWZLPb15	0.54	3.31	0.60	42.0
SWZLPsz16	0.54	3.14	0.57	23.3
SWZLPsz17	0.54	3.48	0.63	20.6
SWZLPsz18	0.54	4.17	0.75	19.7

**Table 5 materials-12-04197-t005:** Load capacity of beams before and after reinforcement.

Beam Designation	Average Cross-Sectional Dimension of Beams, (m)		Support Spacing, (m)	Destructive Force, (kN)	The Moment of Destruction, (kNm)	Bending Stress at Damage, (MPa)	Deflection (mm)
	Width	Height					
BD5	0.208	0.256	4.5	89.62	67.215	29.585	44.43
Repair BD5	0.208	0.256	4.5	71.40	53.55	23.570	46.75
BD6	0.212	0.245	4.5	78.49	58.871	27.758	48.36
BD6 Repair 1	0.212	0.245	4.5	45.47	34.10	16.079	28.45
Repair 2	0.212	0.245	4.5	71.43	53.57	25.260	48.04

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
