# Peer review of "Old Timber Reinforcement with FRPs"

_materials, 2019, doi:10.3390/ma12244197_

Round 1
Reviewer 1 Report
Manuscripts describe the reinforcement of 130 y old wood. Reinforcement is an essential technique for preservation of historic structures. There are several methods on how to address this issue. The authors are describing one of the possible solution. Manuscript is not organised traditionally. I do not like this idea. Materials and results are mixed in the same chapter. I prefer materials and methods organised in one chapter. I suggest the authors to rewrite the paper, to fully follow the journal guidelines.
What is the novelty? Reinforcement is old technique. Glass fibres are used for reinforcement for decades? Same as epoxy. It is obvious that these fibres improves the mechanical properties of wood?
Title and general comment. 130 y old wood is hardly considered antique? 130 y is not a vast age for the building, at least not for Europe.
Introduction is rather complicated. In the first part, there are quite some general statements. Generalisation of these statements is not the best idea, as there are too many exemptions. Besides, as the larch is one of the prime materials in the respective manuscript, authors should describe at least the fundamental importance of this wood in historical buildings.
Line 26, I suggest the authors to use term biological degradation instead of biological corrosion.
Lines 27-29, I have a problem to follow the red line through the first sentences. The respective properties mentioned depends on various properties, and much depends on the durability and application of wood preservatives and wood modification. Moisture performance is very complex phenomena, and relations between are much more complicated than indicated in the respective sentence. I recommend the authors to rewrite the respective sentences and be as accurate as possible. I suggest avoidance of general statements. Be specific, clearly state which wood species you have in mind.
Lines 33-34. Clearly state, why reinforcement described in lines 32-33, is considerably different from reinforcement of wood described in lines 33-34.
Line 67, please define aggressive environment. This likely depends on the type of material used?
Line 78, please define antique timber. What are the prime reasons for reinforcement? Age? There are reports that age is not a critical factor (see work of Thaler for example). If wood decay fungi attack the wood, reinforcement should address this issue as well.
Chapter 2.
Line 83; can you be sure that the wood from the building is the original one? There are numerous reports about the non-documented replacement of wood in old buildings. Nobody makes a drama 100 y ago if particular wood needs to be replaced. Moreover, think about WW2, when many cities were severely damaged. Did you perform dendrochronological analysis? What was the location of the building, city, region, climate conditions? Was house inhabited or not? What is the origin of wood? Did you notice any signs of biological degradation?
Line 90, Describe the conditioning (duration, parameters ...)
Results 2.1
Table 2 is challenging to understand. Authors have to organize text to make table 2 on one side only. There is no need for double space. There is no need to report density with two decimals accuracy. The mistake is far above this range. What is the meaning of abbreviations: Kw, USM, USC… what is spring wood and summer wood?
Table 3, authors should report MoR, and MoE in MPa and GPa. This are standard units that enable comparison.
Line 178 Paragraph 3,
Why part of the experiment was performed on Larch, and the other part on Scots pine?
Information about age, potential damages are missing.
I miss proper discussion. Results are not compared to the reference ones. There has been quite some similar manuscripts published?
Why the beams tested requires reinforcement?
What was the reason for the renovation of the building?
In the respective paper, only static tests were performed. How to predict long term dynamic performance?
What will happen if MC increases? How reinforcement interact with present/potential fungal degradation?
I would like to see the comparison of the performance of reinforced wood with “virgin” wood.
Considerable part of the references about reinforcement is not considered (e.g. work of Subic). There are quite some citations coming from one of the coauthors.
Author Response
Dear Reviewer,
Thank you for your valuable comments.
Manuscripts describe the reinforcement of 130 y old wood. Reinforcement is an essential technique for preservation of historic structures. There are several methods on how to address this issue. The authors are describing one of the possible solution. Manuscript is not organised traditionally. I do not like this idea. Materials and results are mixed in the same chapter. I prefer materials and methods organised in one chapter. I suggest the authors to rewrite the paper, to fully follow the journal guidelines.
Thank you for your correct and critical comments. Corrections have been made to improve the readability of the article.
What is the novelty? Reinforcement is old technique. Glass fibres are used for reinforcement for decades? Same as epoxy. It is obvious that these fibres improves the mechanical properties of wood?
Of course, fiber composite reinforcements have been known for a long time. And carbon fibers have been used for about 30 years. However, these fibers are sporadically used and poorly recognized in the applications of wooden constructions as opposed to reinforced concrete constructions. However, the novelty on the market is the use of basalt fibers to strengthen wooden structures. We confirm the reviewer's opinion that it is obvious that these fibers improve the mechanical properties of wood, but to be able to use them optimally, it is necessary to test the effectiveness of reinforcements using different fibers. Therefore, these studies are continued and the results published.
Title and general comment. 130 y old wood is hardly considered antique? 130 y is not a vast age for the building, at least not for Europe.
Thank you for your attention. In fact, in many cases wood over one hundred years old is not considered antique in Europe. In Polish conditions, however, it is often treated as a monument, although it is not formally it. However, we divide the reviewer's attention and correct the article title for " Old timber reinforcement with FRPs ".
Introduction is rather complicated. In the first part, there are quite some general statements. Generalisation of these statements is not the best idea, as there are too many exemptions. Besides, as the larch is one of the prime materials in the respective manuscript, authors should describe at least the fundamental importance of this wood in historical buildings.Line 26, I suggest the authors to use term biological degradation instead of biological corrosion.Lines 27-29, I have a problem to follow the red line through the first sentences. The respective properties mentioned depends on various properties, and much depends on the durability and application of wood preservatives and wood modification. Moisture performance is very complex phenomena, and relations between are much more complicated than indicated in the respective sentence. I recommend the authors to rewrite the respective sentences and be as accurate as possible. I suggest avoidance of general statements. Be specific, clearly state which wood species you have in mind.Lines 33-34. Clearly state, why reinforcement described in lines 32-33, is considerably different from reinforcement of wood described in lines 33-34.
Regarding the comments, from the beginning of chapter 1 to line 34 the reviewer's suggestions were implemented and the article was significantly rebuilt.
Line 67, please define aggressive environment. This likely depends on the type of material used?
Good point. In the article, we removed the controversial sentence.
Line 78, please define antique timber. What are the prime reasons for reinforcement? Age? There are reports that age is not a critical factor (see work of Thaler for example). If wood decay fungi attack the wood, reinforcement should address this issue as well.
Good point. The sentence was rebuilt and the analyzed issues were clarified in the text.
Chapter 2.
Line 83; can you be sure that the wood from the building is the original one? There are numerous reports about the non-documented replacement of wood in old buildings. Nobody makes a drama 100 y ago if particular wood needs to be replaced. Moreover, think about WW2, when many cities were severely damaged. Did you perform dendrochronological analysis? What was the location of the building, city, region, climate conditions? Was house inhabited or not? What is the origin of wood? Did you notice any signs of biological degradation?
Based on the local vision and the interview with the person who had used these objects for over 60 years, there was no reason to doubt that they were not originally built-in beams. Therefore, no dendrochronological tests were performed. For the purposes of the research, it was not important whether these beams are 100 or 150 years old, but whether it is possible to strengthen old wood used for several decades in the structure. These facilities were not residential buildings but facilities of brick factories. The wood used for testing was healthy without any signs of biological corrosion.
Line 90, Describe the conditioning (duration, parameters ...)
The article details the conditioning of measurements.
Results 2.1
Table 2 is challenging to understand. Authors have to organize text to make table 2 on one side only. There is no need for double space. There is no need to report density with two decimals accuracy. The mistake is far above this range. What is the meaning of abbreviations: Kw, USM, USC… what is spring wood and summer wood?
We agree with a good point. The table has been corrected. KW, KS, KG abbreviations have been translated in the text: KW - exclusive class, KS - medium quality class, KG - lower quality class, and “rejection”.
USM and USC have been translated in the table title: USM - marginal zone knotting index; USC - general knotting index.
In Poland, the terms interchangeably are used: spring wood and summer wood. The text introduces terms more commonly used in Europe.
Table 3, authors should report MoR, and MoE in MPa and GPa. This are standard units that enable comparison.
Good point. During the test, the test apparatus did not allow us to read continuously. Therefore, MOE designation was difficult. Therefore, it was limited to stating the load capacity of the element.
Line 178 Paragraph 3,
Why part of the experiment was performed on Larch, and the other part on Scots pine?
The article presents the results of observations on the issues of strengthening / repairing old wood carried out at two different universities. Therefore, there are two different species of wood in the article.
Information about age, potential damages are missing.
The age of the wood used in the research presented in chapter 3 is given in the first sentence of chapter 3. It was 130-year-old wood. The description of the technical condition of the beams is detailed in chapter 3.
I miss proper discussion. Results are not compared to the reference ones. There has been quite some similar manuscripts published?
There are few examples of destructive testing using old wood in the literature. The article supplements the discussion of the results.
Why the beams tested requires reinforcement?
Reinforcement efficiency was analyzed on the harvested wood, and the beams themselves did not require this reinforcement or repair. They were only research material.
What was the reason for the renovation of the building?
Test items were obtained from:larch - from the liquidated brick factory,pines - renovation of the building with replacement of all ceilings.
In the respective paper, only static tests were performed. How to predict long term dynamic performance?
In the presented paper, tests were carried out under static load. Our other research (long-term and repeatedly variable load [27]) shows that reinforcing wood using fiber composites behaves well under long-term and repeatedly variable loads. The tests did not reveal the flow of glue, as well as the effect of repeated freezing and thawing on the load capacity of the joint.
What will happen if MC increases? How reinforcement interact with present/potential fungal degradation?
These relationships were not analyzed in these studies.
I would like to see the comparison of the performance of reinforced wood with “virgin” wood.
The results of tests for unreinforced larch wood are presented in Table 3, and for reinforced larch wood in Table 4. However, the increase in the load capacity of reinforced wood in relation to unreinforced wood is shown in Figure 4.
Comparison of the effectiveness of pine beam repairs (test BD5 - repair 1, test BD6 - repair 1 and repair 2) compared to the original load capacity of BD5 and BD6 beams determined in the destructive test in an analogous scheme (test up to the first signs of destruction) is shown in Fig. 6 and table 5.
Considerable part of the references about reinforcement is not considered (e.g. work of Subic). There are quite some citations coming from one of the coauthors.
Thank you for your attention. I will try to reduce autocytes in subsequent articles. This article touches on various threads, which is why I have referred to so many works.
Thank you very much for the review.
Please find attached the article with the changes.
Thank you.

Reviewer 2 Report
This paper presents findings from tests carried out on wood beams reinforced with FRPs. The following items are to be addressed before the manuscript can be published:
Line 12, the abstract mentions the use of carbon and aramid fibers but only BFRP and GFRP were used. It seems that glass fibers were also used? What were the properties of the cured FRPs? Did the authors carry out any observational or material testing on the antique wood? for example, was there any noticeable rotting? Cross sectional losses? Damage? Was the epoxy used specific for wood? Or was it for general use (i.e. concrete)? The manuscript is very short. Perhaps the authors can add a section on the above comment. The authors are also advised to enhance the introduction section. This can be easily done by describing previous works such as the following or any other that they may come across. https://doi.org/10.1016/j.culher.2012.01.016 https://doi.org/10.1016/j.engstruct.2019.109542 http://hdl.handle.net/10902/12378 In Fig. 12, why only one side of the beam was wrapped? Oftentimes, beams are wrapped at the edges to minimize edge debonding. The first and fifth conclusions are similar. One of them can be deleted. Additionally, some of the conclusions could be compiled into one point (i.e. 1 and 3).
Author Response
Dear Reviewer,
Thank you very much for your valuable comments.
Line 12, the abstract mentions the use of carbon and aramid fibers but only BFRP and GFRP were used. It seems that glass fibers were also used? What were the properties of the cured FRPs? Did the authors carry out any observational or material testing on the antique wood? for example, was there any noticeable rotting? Cross sectional losses? Damage?
Thank you very much for a good comment.
The summary describes that mats and strips made of carbon and aramid fibers, BFRP basalt rods and glass GFRP were used in the study. And all these tests were carried out on these FRP composite materials. All properties of FRP materials are described in the introduction.
It is difficult to refer to cross-sectional losses, because we do not know the primary dimensions, there was slight biological degradation on the wood. There was no surface corrosion on the surface, the beams were in good condition.
Was the epoxy used specific for wood? Or was it for general use (i.e. concrete)?
Epoxy resin for general use was used in the studies.
The manuscript is very short. Perhaps the authors can add a section on the above comment. The authors are also advised to enhance the introduction section. This can be easily done by describing previous works such as the following or any other that they may come across. https://doi.org/10.1016/j.culher.2012.01.016 https://doi.org/10.1016/j.engstruct.2019.109542 http://hdl.handle.net/10902/12378
Thank you very much for the above suggestions. The introductory part has been modified. However, it is difficult to describe all the issues in one article, some methods are very debatable, so in order not to complicate the topic, it was decided to give citations instead of describing the given work.
In Fig. 12, why only one side of the beam was wrapped? Oftentimes, beams are wrapped at the edges to minimize edge debonding.
Beam wrapping was done on one side only, because there was damage (fracture) resulting from destructive tests performed earlier. The necessity to use a band (wrapping resulted from various displacements of damaged wood fragments, which caused premature delamination of the wood-CFRP joint. The use of the band prevented this phenomenon in another attempt to repair this beam.
The first and fifth conclusions are similar. One of them can be deleted. Additionally, some of the conclusions could be compiled into one point (i.e. 1 and 3).
The first and third applications have been put together into one. The fifth application was withdrawn because it is similar to the first one.
Please find attached the corrected article.
Thank you very much for your valuable comments.

Round 2
Reviewer 1 Report
Authors have revised the manuscript. Changes are evident in the manuscript as well as in the provided "response". Authors have discussed all of the comments. Comments are more or less considered. I found the manuscript considerably improved. Some of the comments are challenging to address within the current manuscript, and they were meant as a suggestion for future research work and research planning.
As I do not have additional comments, that can be addressed within the current frame of the manuscript, I recommend the editors, to accept the manuscript in the current form.
Reviewer 2 Report
thank you for your effort.